# Impact of Nisin-Producing Strains of *Lactococcus lactis* on the Contents of Bioactive Dipeptides, Free Amino Acids, and Biogenic Amines in Dutch-Type Cheese Models

**DOI:** 10.3390/ma13081835

**Published:** 2020-04-13

**Authors:** Monika Garbowska, Antoni Pluta, Anna Berthold-Pluta

**Affiliations:** Division of Milk Technology, Department of Food Technology and Assessment, Institute of Food Sciences, Warsaw University of Life Sciences-SGGW, Nowoursynowska 159c Street, 02-776 Warsaw, Poland; antoni_pluta@sggw.pl (A.P.); anna_berthold@sggw.pl (A.B.-P.)

**Keywords:** biogenic amine, free amino acids, additional culture, L-carnosine, anserine

## Abstract

The goal of this study was to determine changes in contents of free amino acids, biogenic amines, and bioactive dipeptides (anserine and L-carnosine) in cheese models produced with the addition of nisin-producing strains of *Lactococcus lactis* over their ripening period. After 5 weeks of ripening, contents of total biogenic amines in the cheese models with the addition of *L. lactis* strains were lower than in the control cheese model. The cheese models examined differed significantly in contents of free amino acids through the ripening period. Individual free amino acids, such as ornithine, were found in some of the cheese models, which is indicative of their specific microbial activities. Both anserine and L-carnosine were detected in all variants of the cheese models. After 5-week ripening, the highest total content of bioactive dipeptides was determined in the cheese models produced with the nisin-producing culture of *L. lactis* 11454 (142.15 mg∙kg^−1^).

## 1. Introduction

Proteolysis is one of the major biochemical processes of cheese ripening and contributes to taste and texture development [1]. Firstly, casein is hydrolyzed to a number of medium-sized peptides which are next degraded to shorter peptides and amino acids. These transformations lead to the accumulation of free amino acids, including precursors of biogenic amines (BAs). BAs are low-molecular-weight nitrogenous compounds that are formed in foodstuffs by microbial decarboxylation of the precursor amino acids. They may be found detrimental considering their neuroactive activities, and hence their levels in food products should be under strict control. Their presence in foodstuffs may induce multiple health, issues especially for sensitive persons, that are manifested by, e.g., headaches, vertigos, nauseas, vomiting, and increased arterial blood pressure [2,3]. According to the European Food Safety Agency (EFSA) [4], cheeses represent fermented food products most often associated with the presence of Bas; however, there are no EU regulations that set the upper level of BAs in these products. BAs have been documented to occur in different varieties of cheeses produced from cow, sheep, or goat milk. Most of the food products fermented by lactic acid bacteria (LAB), including cheeses, contain trace amounts of histamine, tyramine, putrescine, cadaverine, and 2-phenylethylamine, which are products of the decarboxylation of histidine, tyrosine, ornithine, lysine, and phenylalanine, respectively [5,6,7,8,9,10,11,12]. In cheese, putrescine is produced mainly by deamination of agmatine. The presence of BAs in cheeses made from pasteurized milk is associated with nonstarter lactic acid bacteria (NSLAB) [13], which exhibit both a high capability for BA production as well as resistance to heat treatment [14].

Nisin is a polypeptide bacteriocin produced by certain strains of *Lactococcus lactis*, which has no or weak activity against Gram–negative bacteria, mold, and yeast but is effective against Gram-positive bacteria, including pathogenic bacteria (*Staphylococcus aureus*, *Listeria monocytogenes*, and *Clostridium botulinum*) [15]. It is used in the food industry as a preservative (code E234). In the European Union, nisin is allowed to be used for clotted creams, unripened cheeses (only mascarpone), ripened and processed cheeses, heat-treated meat products, pasteurized liquid eggs, and desserts like semolina and tapioca puddings [16].

The use of bacteriocin-producing strains to induce the lysis of LAB is a relatively new way to accelerate cheese ripening. This solution brings many benefits. Firstly, there are no legal barriers that occur when using nisin as a food additive [17]. Secondly, there is no need for any additional technological operations or equipment. In addition, nisin is distributed uniformly in the cheese matrix. At the same time, the presence of bacteriocin can ensure the safety and hygienic quality of the product.

LAB peptidases are intracellular enzymes, and therefore bacterial lysis may play a major role in accelerating cheese maturation by favoring the access of enzymes to their substrates [18,19,20,21,22]. The use of the nisin-producing strain of *L. lactis* subsp. *lactis* allowed to limit the content of biogenic amines in cheeses produced from raw goat’s milk [20], which suggests that bacteriocinogenic cultures can also be used to limit the formation of undesirable ingredients in cheese. Rossi and Veneri [21] show that bacteriocinogenic cultures had no inhibitory effect on the NSLAB group, which plays a significant role in the maturation and production of typical cheese sensory traits. In turn, research on the use of strains producing bacteriocins in the production of Cheddar cheeses to limit the development of NSLAB responsible for causing defects in these cheeses [23] shows the effectiveness of using these strains.

The main problem with the use of bacteriocin-producing strains in fermented foods, including cheese, is in situ antimicrobial efficacy, which can be adversely affected by various factors such as bacteriocin binding to food ingredients, their deactivation by proteolytic enzymes, and the chemical and physical characteristics of food (e.g., pH, NaCl, fat) [15,24]. Due to these limitations, knowledge about the effect of additional bacteriocinogenic strains of lactic bacteria on the SLAB and NSLAB groups during the production and maturing of cheeses, as well as on the quality characteristics of these products, seems insufficient.

Apart from having the basic nutritional value, certain food products contain substances without specifically established nutraceutical functions but which may significantly affect human health [25]. Such bioactive substances are currently in the focus of interest of the food and pharmaceutical industries. Recent investigations have confirmed the presence of some bioactive nonproteinaceous amino acids in cheeses, e.g., gamma-aminobutyric acid (GABA) and ornithine (Orn) [26,27]. Kurata et al. [28] demonstrated sedative and hypnotic effects of L-ornithine in rats exposed to severe stress. Ornithine was also shown to alleviate fatigue by increasing energy consumption effectiveness and to facilitate ammonia excretion [29]. Other bioactive substances, like L-carnosine (β-alanyl histidine), a dipeptide composed of α-alanine and L-histidine and its methylated analogue anserine (1-methyl carnosine) have been detected in skeletal muscles and nervous tissues of different vertebrates like fish, birds, and mammals. Their contents in a human body depend on gender (higher contents are reported in men), age (contents decrease with age), and diet (lower levels of L-carnosine and anserine are determined in vegetarians) [30]. L-carnosine synthesis in skeletal muscles is determined by the availability of amino acids, i.e., alanine and histidine, in the body. It exhibits multiple beneficial activities including antiglycating, antioxidative, and anti-crosslinking ones [30,31]. As these activities are associated with the ageing process, L-carnosine is hypothesized to be an endogenous neuroprotective agent counteracting this process. Some investigations suggest that diet supplementation with carnosine may be effective in the prevention or treatment of neurodegenerative diseases such as Parkinson’s or Alzheimer’s diseases. In addition, L-carnosine is capable of chelating metal ions (iron, copper, zinc, and cobalt) and of reducing their toxicity [30,32]. Diet supplementation with L-carnosine and anserine has proved effective in reinforcing the cognitive functions in the elderly [33]. The mechanism of carnosine action has not been explicitly recognized yet, but it presumably improves functions of the nervous system in temporal lobes and frontal lobes of the cerebral cortex [34]. Some studies have demonstrated L-carnosine to be a natural inhibitor of angiotensin convertase enzyme and thus to be able to affect arterial blood pressure regulation, to exhibit antiproliferative properties, and to influence functions of the cardiovascular system [30,35,36]. Tomonaga et al. [37] demonstrated that oral administration of a chicken breast extract increased L-carnosine level in the brain. Results of the above-discussed investigations indicate that analyses of L-carnosine and anserine contents in food products are indispensable to develop an appropriate supplementation therapy. Scientific investigations concerning L-carnosine presence in foodstuffs refer mainly to meat products. However, no data have been found in the available literature regarding the presence of these dipeptides in cheeses.

In view of the above, a study was undertaken with the aim to determine the composition of free amino acids (including ornithine) as well as contents of biogenic amines and bioactive dipeptides (L-carnosine and anserine) in cheese models produced with the addition of nisin-producing strains of *L. lactis.* Amino acids and biogenic amines in these cheese models were characterized by liquid chromatography coupled with electrospray ionization ion trap tandem mass spectrometry (LC-ESI-IT-MS/MS).

## 2. Material and Methods 

### 2.1. Experimental Material

The experimental material included four model cheeses produced with a CHN-19 (Chr. Hansen, Czosnów, Poland) culture (*L. lactis* ssp. *cremoris*, *L. mesenteroides* ssp. *cremoris*, *L. lactis* ssp. *lactis biovar diacetylactis*) used as the basic starter and with *Lactococcus lactis* PCM 476 (does not produce nisin), nisin-producing *Lactococcus lactis* PCM 2379 (Polish Collection of Microorganisms of the L. Hirszfeld Institute of Immunology and Experimental Therapy of the Polish Academy of Sciences in Wrocław, Poland), and nisin-producing *Lactococcus lactis* ATCC 11454 used as adjunct cultures, one for each of three cheese variants. All the organisms were activated from their frozen forms by one transfer into sterile reconstituted skim milk (SRSM) supplemented with glucose and yeast extract and incubation at 30 °C for 20 h. This was followed by two transfers with 1 mL of inoculum into 100 mL sterile reconstituted skim milk (RSM). Subsequently, 1 mL of the inoculum from the SRSM were transferred into 100 mL of RSM. During the activation, all organisms were incubated at 30 °C. Prepared in this way, LAB was used as adjunct cultures in the preparation of cheese models. The additional cultures used in cheese model manufacture were selected based on their good technological aptitude and proteolytic activity. 

### 2.2. Preparation of Cheese Models

Cheese models were prepared in sterile centrifuge bottles (500 mL) (Nalgene, Thermo Scientific, Warsaw, Poland) according to the method described in our previous study [38]. 

Four variants of cheese models were prepared in the study as shown in Table 1. 

### 2.3. Chemical Composition

Water, fat, and protein content was determined using the FoodScan TM2 analyzer (Foss, Poland). Cheese pH was measured at 20 °C using a Portamess 900 pH-meter by Knick. All determinations were conducted after 5 weeks of ripening.

### 2.4. Determination of Free Amino Acids, Biogenic Amines, and Bioactive Dipeptides in Cheese Models with LC-ESI-IT-MS/MS

Determination of free amino acids, biogenic amines, and bioactive dipeptides in cheese models with LC-ESI-IT-MS/MS was conducted according to the methodology described by Szterk and Roszko [39]. A quantity of 1.2 g of cheese was weighed into falcon test tubes, then 5 mL of a dichloromethane–methanol mixture (2:1) was added (Sigma-Aldrich, Poznań, Poland) and the sample was homogenized in a UP200S ultrasound homogenizer (Hielscher Ultrasonics GmbH, Teltow, Germany) at the amplitude of 60% for 1.3 min to separate fat. Afterwards, 5 mL (40 mM) of lithium carbonate (Sigma-Aldrich, Poznań, Poland) was added to the sample, which was again homogenized at 60% of the maximum power of the sonifier for 1 min for amino acid extraction. After the extraction, the samples were centrifuged at 10,000*g* and a temperature of 25 °C for 10 min, and the upper phase was transferred to one-mark flasks and filled with lithium carbonate up to the volume of 25 mL. Next, 2 mL of the resultant solution was transferred to reaction bottles, to which 0.05 mL of an internal standard (170 mg of N-butylamine, Avantor Performance Materials, dissolved in 10 mL of 40 mM lithium carbonate, pH 11), 3 mL of acetonitrile (Sigma-Aldrich, Poland), and 0.8 mL of dansyl chloride (Sigma-Aldrich, Poland) were added. Amino acids were derivatized at a temperature of 60 °C for 30 min. After incubation was complete, 0.08 mL of pyridine (Sigma-Aldrich, Poland) was added to the solution to enable decomposition of the excess amount of dansyl chloride. Next, the solution was stirred and left for 10 min. Then, 3 mL of saturated brine was added to modify the phase separation coefficient and to enable acetonitrile precipitation. After mixing, the samples were centrifuged at 1500*g* and a temperature of 25 °C for 2 min. The upper phase was collected in a round-bottom flask, whereas reaction bottles were filled with ethyl acetate (Sigma-Aldrich, Poland) and the resultant solution was mixed and centrifuged at 1500*g* and 25 °C for 5 min. Afterwards, the upper phase was transferred to a round-bottom flask containing the previously collected fraction. The solvent was evaporated to dryness in a rotary evaporator (Büchi Labortechnik AG, Warsaw, Poland) at a temperature of 50 °C and under the pressure of 102 mBa. The residue was dissolved in 5 mL of methanol (Sigma-Aldrich, Poland). Then, 2 mL of the solution was filtered through Teflon syringe filters (hydrophobic PTFE, 0.22 μm, 13 mm) to test tubes inserted into an autosampler. In the chromatographic analysis, the sample injection volume was 5 μL. 

Amino acids, biogenic amine, and bioactive peptides were determined with the Shimadzu LCMS 2020 HPLC system equipped with a DAD detector and a mass spectrometer operated in the ESI (electrospray ionization) mode. Separations were performed on a 5 mm × 2 mm × 250 mm COSMOSIL AR-II column thermostated at 85 °C. Mobile phase flow rate was 600 μL∙min^−1^. The gradient used for separations is described in Table 2.

The mass spectrometer setup was as follows: DL temperature = 200 °C; nebulizing gas flow = 1 L∙min^−1^; heat block temperature = 350 °C; drying gas flow = 15 L∙min^−1^.

In this study, all analyses were done in duplicate.

### 2.5. Statistical Analysis

Results obtained were subjected to a statistical analysis using Statistica version 13 software (TIBCO Software Inc., Kraków, Poland, 2017). One-way analysis of variance (ANOVA) was conducted. Tukey test was applied to compare the significance of differences between mean values at a significance level of α = 0.05.

## 3. Results and Discussion

### 3.1. Chemical Analysis of Cheese Models

Table 3 summarizes the effect of adjunct *L. lactis* strains on the physicochemical composition of model cheeses. No differences (*p* < 0.05) were observed in the main parameters of moisture, protein, fat, moisture on fat-free basis (MFFB), and fat in dry matter (FDM) in model cheeses, compared to control cheeses. The percentage contents of fat and water in cheese models were fitted within ranges of 18.03–18.50% and 50.31–51.02%, respectively. The prepared cheese models were similar in terms of composition to Dutch-type cheese. 

The content of protein in cheese models ranged from 25.31% to 26.04% (Table 3). The content of total protein depends mainly on contents of fat and water and reaches ~26% in semihard cheeses. Reduced-fat cheeses are, however, characterized by a considerably higher protein content and harder structure compared to full-fat cheeses [40,41].

### 3.2. Contents of Free Amino Acids in the Analyzed Cheese Models

Contents of individual free amino acids (FAAs) and total free amino acids (TFAAs) in Dutch-type cheese models manufactured with different cultures during their ripening are shown in Table 4. The cheese models examined were characterized by high variability in FAA contents over the ripening period. Threonine and cystine were not detected in any of the cheeses. In general, the most abundant amino acid, regardless of the ripening time and the culture used, was asparagine. The lowest contents were determined for alanine, arginine, glutamine, glycine, methionine, serine, tryptophan, taurine, and citrulline, whereas medium ones were found for isoleucine, leucine, phenylalanine, proline, and valine. In turn, contents of aspartic acid, glutamic acid, histidine, lysine, tyrosine, sarcosine, and ornithine differed the most throughout the ripening period of the analyzed cheeses. 

Free amino acid composition has been evaluated in several cheese varieties. The content of FAAs in cheese is related to the manufacturing technology (type of curd, addition of proteinases, starters, ripening conditions), duration of ripening, and the extent and type of proteolysis [6,9,10,26,42,43]. 

Degradation of milk proteins and an increase in FAA content depend, to a large extent, on the activity of microorganisms (lactococci in particular) and on their enzymatic system released after cell lysis. Proteolytic enzymes of microbiota are responsible for debranching the terminal peptidic bonds in proteins and peptides, which results in FAA release. Also of importance are successive transformations of FAAs and synthesis of other products, e.g., substances with sensory activities or biogenic amines [44].

Significant differences were determined in the contents of individual FAAs in all four variants of the cheese models. In the case of control cheeses, no glutamine, serine, or tyrosine were detected throughout the ripening period, whereas the highest content was reported for histidine (123.91 mg∙kg^−1^ after 5 weeks of ripening). After the same period, the cheese models manufactured with nisin-producing *L. lactis* 11454 had low contents of arginine, glycine, methionine, serine, taurine, and citrulline, which ranged from 10.47 to 33.87 mg∙kg^−1^. Slightly higher contents were determined for the following amino acids: alanine, aspartic acid, glutamine, isoleucine, leucine, lysine, phenylalanine, and ornithine (from 33.27 to 75.29 mg∙kg^−1^ after 5 weeks). In the case of the cheese models produced with the addition of *L. lactis* 11454, high contents were assayed for valine, histidine, proline, asparagine, and glutamic acid, which after the 5-week ripening accounted for 107.30, 117.86, 130.96, 187.57, and 428.93 mg∙kg^−1^, respectively. During 5 weeks of ripening, the cheese models manufactured with nisin(+) *L. lactis* 2379 were characterized by low contents of alanine, arginine, glutamine, glycine, histidine, lysine, methionine, serine, and taurine. In turn, higher contents were determined for aspartic acid, isoleucine, leucine, phenylalanine, proline, tyrosine, valine, citrulline, and ornithine. In these cheese models, higher contents were assayed for asparagine and glutamic acid, which accounted for 140.71 and 706.76 mg∙kg^−1^, respectively after 5 weeks of the ripening period. The cheese models containing the adjunct culture of *L. lactis* 476 were characterized by slightly different contents of amino acids compared to the cheese models produced with the other two nisin(+) *L. lactis* strains used in our study. Over the ripening period, glutamine, serine, and tyrosine were not found in these cheese models or in the control ones. In addition, the cheese models had low contents of alanine, arginine, aspartic acid, glycine, histidine, lysine, methionine, proline, tryptophan, taurine, and citrulline over the entire period of ripening. In turn, very high contents, maintained at similar levels throughout the ripening period, were found for isoleucine and leucine, these being 85.67 and 110.01 mg∙kg^−1^, respectively after 5 weeks. After this period, leucine content in these cheese models was the highest among all models tested. Leucine is classified as an essential amino acid which enables determining the degree of proteolysis. Leu, Phe, and Val are released mainly upon proteolysis of α_S1_-casein (which is rich in these amino acids), which is more extensive than the proteolysis of β-casein [45].

After 5 weeks of ripening, the content of ornithine in the analyzed cheese models ranged from 35.86 to 67.81 mg∙kg^−1^. Its highest content in the examined period was determined in the cheese models containing an adjunct culture of *L. lactis* 11454, which seems interesting considering its confirmed bioactive functions, e.g., its sedative effect [28]. Diana et al. [26] reported an ornithine content range of 450 ± 60 mg∙kg^−1^, showing that ornithine concentration was higher (*p* < 0.05) in cheese made from cow’s milk than that found in cheese made from goat’s milk. In turn, Renes et al. [27] found high average concentrations of ornithine (2355.76 mg/kg) in all sheep milk cheese variants at 240 days of ripening.

Glutamic acid, responsible for the intensification of cheese flavor, was found in high amounts in the cheese model with the addition of *L. lactis* 2379. Its high content has been related to enhancing flavor in Kefalograviera cheese [46]. Arginine—which is responsible for the development of bitter taste in cheeses—was found in low amounts in all variants of cheese models analyzed in our study. A high content of proline was assayed in the cheese model with the addition of *L. lactis* 11454 during the ripening time as compared with the other of cheese models. This may be found beneficial, as proline imparts sweet taste to cheeses and is especially desirable in Emmental type cheeses [47].

Ripening is relatively expensive process, and the use of additional culture in cheese production could reduce the ripening time of cheese, providing technological benefits [48]. In many cheese varieties during ripening, the content of the total free amino acids (TFAAs) increased. In this study contents of TFAAs were significantly (*p* < 0.05) affected by the adjunct culture. In model cheeses, they reached 447.94, 574.22, and 800.92 mg∙kg^−1^ after 1, 3, and 5 weeks of ripening, respectively. In turn, in the cheese model with the addition of nisin-producing *L. lactis* 11454, TFAA content was almost twofold higher than in the control cheeses and after 5 weeks reached 1592.78 mg∙kg^−1^. In the cheese models containing nisin(+) *L. lactis* 2379, TFAA content reached 1099.52 mg∙kg^−1^ after 1 week and increased to 1437.59 mg∙kg^−1^ after 5 weeks of ripening. The lowest total content of free amino acids was determined in the cheese model containing *L. lactis* 476, and it was correlated with the lowest total content of BA in these models. Perhaps this is due to interactions between *L. lactis* 476 and LAB strains of the basic starter culture (CHN-19) used in the cheesemaking process. Lysis of the SLAB cells due to the activity of nisin-producing strains could have influenced the higher TFAA content in cheese models with the addition of *L. lactis* 11454 and *L. lactis* 2379 strains by the release of intracellular peptidases. Sallami et al. [49] observed that use of a nisin Z-producing *L. lactis* subsp. *lactis biovar diacetylactis* UL719 significantly increased the content of free amino acids in Cheddar cheese. The presence of *L. diacetylactis* UL719 accelerated autolysis of lactobacilli and proteolysis [49].

### 3.3. Contents of Biogenic Amines in the Analyzed Cheese Models

Contents of biogenic amines (BAs) in the cheese models examined were found to differ significantly and after 5 weeks of ripening ranged from 32.48 mg∙kg^−1^ in the cheese models with *L. lactis* 476 to 53.41 mg∙kg^−1^ in the control cheese model (Table 5). After 5 weeks, however, the highest total BA content was determined in the control cheese models (53.41 mg∙kg^−1^). This indicates that the adjunct starter cultures did not contribute to an increase in BA content but rather to its lower increase after 5 weeks of ripening. No agmatine or phenylethylamine were detected in any of the cheese models throughout the ripening period. Differences were also noted in the various cheeses, and they were mainly due to the ripening time of cheeses and to adjunct LAB cultures used to manufacture them, which could promote the growth of proteolytic microbiota and, by this means, BAs synthesis during ripening [11].

In general, no correlation was found between contents of TFAAs and BAs in the cheese models, although both control cheese and LC 11454 cheese, which had the higher amount of histidine, showed the higher amount of histamine too. The samples having the highest content of TFAAs were not always characterized by the highest content of BAs. For example, the control cheese model with the highest total BA content (53.41 mg∙kg^−1^) had at the same time one of the lowest contents of TFAAs (800.92 mg∙kg^−1^) after 5 weeks of ripening. In turn, the cheese model containing a nisin(+) *L. lactis* 11454 had the highest content of TFAAs among all models tested after 5 weeks of ripening but contained ca. 10 mg∙kg^−1^ BA less compared to the control cheese model. These results confirm dependencies described by other authors [9] and may be affected by other environmental factors like pH, salt concentration, and temperature. These factors may determine the activity of decarboxylating enzymes and may have a stronger impact on BA synthesis than the availability of a precursor free amino acid synthesized during the ripening process [50]. At the first stage of ripening of the cheese models, BA content was observed to increase, whereas after 5 weeks of ripening its decreased value was determined in the cheese models produced with the addition of *L. lactis*; this decrease could be due to utilization of the precursor free amino acids by the adjunct starter cultures.

Histamine and tyramine are described as the most abundant biogenic amines in cheeses made of cow and sheep milk [9,43]. The amount of histamine found in the Herby cheese samples tested by Ekici et al. [51] ranged from <0.033 to 469.00 mg∙kg^−1^. Sagun et al. [52], reported that the concentration of histamine gradually increased during ripening from 21.90 mg∙kg^−1^ on the first day to 46.20 mg∙kg^−1^ after 3 months. The amounts of tyramine found in the Herby cheeses were up to 725.21 mg∙kg^−1^ [51] and 1125.50 mg∙kg^−1^ [53]. In our study, histamine contents were higher in all cheese models examined compared to tyramine levels. In addition, after 5 weeks of ripening they were significantly (*p* < 0.05) higher in the control cheese model and in the models containing nisin-producing *L. lactis* than in the variant of cheese model without nisin-producing *L. lactis*. From the toxicological perspective, histamine and tyramine are claimed to be the most toxic BAs and are especially significant to the food safety. Their toxicities vary depending on the quantity of BAs ingested and individual sensitivity. Typical symptoms of BA poisoning are migraines, headaches, edemas, increased/decreased blood pressure, skin and food allergies, nausea, vomit, and diarrhea [54,55]. Their safe levels have been established in the European legislation. According to the European Food Safety Agency (EFSA), there have been no adverse health effects after histamine intake in the amount of 25–50 mg per capita per meal and after tyramine intake in the amount of 600 mg per capita per meal [4]. Considering these values and the average amount of cheese consumed per meal, none of the cheese variants analyzed in our study would pose any significant threat to consumer health in terms of BA levels. 

Putrescin, cadaverine, spermidine, and spermine contents were low in all cheese models throughout their ripening period. According to Komprda et al. [56], the mean level of spermine and spermidine in Dutch-type hard cheeses were 0.2 ± 0.1 and 0.3 ± 0.1 mg∙kg^−1^, respectively. In long-ripening cheeses, the level of these biogenic amines is much higher and reached tens of milligrams per kg [11]. High contents of putrescine and cadaverine were detected in overripe cheeses [57] and in cheeses manufactured from raw milk [58]. Both these compounds are responsible for the development of undesirable flavor traits of cheese [59]. The presence of cadaverine in cheeses is associated with the contaminating bacteria, and its very low content in the analyzed cheese models is indicative of a good hygienic quality of the milk used to make them. Tryptamine was not detected in the cheese models with nisin(+) *L. lactis* 11454 during their ripening, whereas in the other variants of cheeses it was determined in very small amounts, ranging from 2.49 mg∙kg^−1^ in the control cheese model after one week to 4.59 mg∙kg^−1^ in the cheese model with the addition of LC 476 after 5 weeks of ripening. Perin et al. [20] observed that when adding a nisin-producing strain *L. lactis* subsp. *lactis* GLc05 to Minas cheeses, the contents of 2-phenylethylamine and cadaverine after 60 days of aging were four times lower than the control cheeses without additional strain. In our study, lower contents of cadaverine, putrescine, tyramine, and total BA were found in cheeses with adjunct *L. lactis* (regardless of the ability to produce nisin) after 5 weeks of ripening in comparison to the control cheese model (Table 5). In our study, we chose to minimize the presence of microorganisms relevant for BA production (NSLAB), so the content of BAs is exclusively determined only by the specific starter and adjunct cultures used. Therefore, it is difficult to compare our results with the literature data in this area.

The analyzed cheese models were characterized by low contents of BAs. It is common knowledge that milk pasteurization is one of the major technological factors which diminish BA accumulation in dairy products, because many of the decarboxylating bacteria die during heat treatment [60]. In our study, the cheese models were manufactured from microfiltered, pasteurized milk (74 °C/15 s); therefore, they should have been expected to have low contents of BAs, derived mainly from the proteolytic activity of the starters used in the cheesemaking process. In addition, the cheese models were small, and hence their ripening period was shorter.

### 3.4. Determination of Anserine and Carnosine in Cheese Models

As in the case of TFAAs, the highest content of total bioactive peptides (TBPs) after 5 weeks of ripening (i.e., 142.15 mg∙kg^−1^) was determined in the cheese models produced with nisin-producing *L. lactis* 11454 (Table 6). These cheese models were also characterized by one of the highest contents of histidine (117.86 mg∙kg^−1^) and the highest contents of alanine (59.42 mg∙kg^−1^) determined after 5 weeks of ripening compared to all other variants of cheese models. This may be found beneficial, because L-carnosine synthesis in human skeletal muscles of depends on the availability of such amino acids as alanine and histidine in the body; therefore, a diet rich in histidine increases carnosine level in muscles. Considering the above, apart from typical nutritional values, this cheese model may exert additional health effects. After 5 weeks of ripening, the highest level of L-carnosine was found in the cheese model containing nisin(+) *L. lactis* 11454. In the other variants of cheeses, L-carnosine was detected mainly in the 1st week of ripening; after 5 weeks, they had a similar content of TBPs (ca. 79 mg∙kg^−1^), which resulted only from the content of anserine. Generally, anserine content remained at a similar and high level in the 1st, 3rd, and 5th weeks of ripening. 

It is unquestionable that the recorded levels of anserine and carnosine in cheese models are low in comparison to those of other foods. In regard to this, Mori et al. [31] identified the content of carnosine and anserine in various foods (meat and fish). They did not detect anserine and carnosine in pork, liver and heart of chicken, scallops, and squids. Considerable amounts of anserine were detected in chicken (2.20–7.11 mg∙g^−1^), beef (0.24–0.84 mg∙g^−1^), and lamb (1.02–1.03 mg∙g^−1^). They observed that relatively high concentrations of carnosine exist in muscles. There is a high probability that carnosine performs crucial biological roles in muscles. Some researchers suggest using carnosine to prevent of neurodegenerative diseases [31,32]. It is unquestionable that the recorded levels are low in comparison of those of other foods.

## 4. Conclusions

The analyzed cheese models produced with the addition of *L. lactis* strains had various total and individual contents of free amino acids. Ornithine was detected in all variants of cheese models, which seems interesting considering its documented bioactive functions. The cheese models were characterized by a low content of BAs. Among BAs, the highest contents were reported for histamine; however, these contents were still several times lower than the level regarded as unsafe for humans. The adjunct LAB cultures used in the cheesemaking process were found to cause a decrease in BA contents in the cheese models after 5 weeks of ripening. The cheese model containing the nisin(+) of *L. lactis* 11454 had the highest contents of TFAAs, ornithine, L-carnosine, and TBPs compared to the other cheese models analyzed in our study. This may point to the interesting properties of *L. lactis* 11454, including its capability to produce bioactive substances during the ripening process of rennet cheeses. It seems that the use of nisin-producing strains in cheesemaking can cause positive effects. These strains increase the amount of free amino acids in cheeses (depth of ripening) and as a result accelerate ripening. In addition, they might contribute to the release of health-promoting compounds in the cheese.

## Figures and Tables

**Table 1 materials-13-01835-t001:** Cheese model variants.

Cheese Model Variant	Coagulating Enzyme	Basic Starter	Additional Starter
Control	Fromase 2200 TL (to all cheese model variants)	2.0% CHN-19 (to all cheese model variants)	—
LC 11454			1.5% *L. lactis* 11454
LC 2379			1.5% *L. lactis* 2379
LC 476			1.5% *L. lactis* 476

Cheese models variants (C, LC 476, LC 2379, LC 11454) were prepared in triplicate.

**Table 2 materials-13-01835-t002:** Gradient used for separations amino acids, biogenic amine, and bioactive peptides.

Time (min)	Composition	
	Mobile Phase A (1% Formic Acid)	Mobile Phase B (20% Methanol in Acetonitrile)
0–15	91%	9%
15–40	50%	50%
40–50	20%	80%
50–60	91%	9% (column washing)

**Table 3 materials-13-01835-t003:** Chemical composition of cheese models after 5 weeks of ripening.

Component	Cheese Model			
	Control	LC 11454	LC 2379	LC 476
Moisture (%)	50.31 a	50.76 a	51.02 a	50.74 a
MFFB (%)	61.37 a	62.11 a	62.60 a	62.06 a
Fat (%)	18.03 a	18.28 a	18.50 a	18.24 a
FDM (%)	36.28 a	37.12 a	37.77 a	37.03 a
pH	5.33 a	5.31 a	5.29 a	5.32 a
Protein (%)	26.01 a	26.04 a	25.96 a	25.31 a

Means with different letters in a line are significantly different (*p* < 0.05, *n* = 6).

**Table 4 materials-13-01835-t004:** Concentrations of individual and total free amino acids (TFAA) in Dutch-type cheese models during ripening (mg∙kg^−1^).

Added	Control			LC 11454			LC 2379			LC 476		
Time of Ripening (week)	1	3	5	1	3	5	1	3	5	1	3	5
Alanine	4.58 a	12.48 AB	11.43 ^AB^	10.91 b	37.05 C	59.42 ^C^	14.52 b	20.10 B	20.25 ^B^	2.51 a	5.38 A	8.44 ^A^
Arginine	18.31 a	17.68 A	16.59 ^A^	18.13 a	16.54 A	17.08 ^A^	19.07 a	19.80 A	15.82 ^A^	16.01 a	17.60 A	17.52 ^A^
Asparagine	73.11 a	77.93 A	88.81 ^A^	133.05 b	136.65 B	187.57 ^C^	126.11 b	135.13 B	140.71 ^B^	69.50 a	80.81 A	93.22 ^A^
Asparagine acid	16.56 b	31.45 B	54.50 ^B^	33.98 c	36.58 B	67.08 ^C^	64.13 d	77.51 C	86.67 ^D^	1.46 a	1.64 A	1.66 ^A^
Glutamine	ND a	ND A	ND ^A^	31.31 b	34.84 B	56.18 ^C^	ND a	ND A	12.57 ^B^	ND a	ND A	ND ^A^
Glutamic acid	10.94 a	11.64 A	30.76 ^A^	132.30 b	177.85 B	428.93 ^B^	524.19 c	534.04 C	706.76 ^C^	2.90 a	3.88 A	35.92 ^A^
Glycine	0.30 a	0.32 A	0.49 ^A^	0.29 a	4.32 C	10.47 ^C^	1.55 a	2.50 AB	3.69 ^B^	0.15 a	0.64 A	1.10 ^A^
Histidine	0.13 a	28.31 C	123.91 ^C^	33.59 c	89.81 D	117.86 ^C^	5.46 a	5.67 A	6.17 ^A^	14.11 b	16.59 B	18.67 ^B^
Isoleucine	84.40 c	77.09 B	70.88 ^B^	68.00 b	70.52 B	75.29 ^B^	51.33 a	58.15 A	59.22 ^A^	66.16 b	69.99 B	85.67 ^C^
Leucine	88.12 c	97.22 C	105.94 ^B^	62.31 ab	58.94 A	57.21 ^A^	50.36 a	55.32 A	59.37 ^A^	71.82 b	79.14 B	110.01 ^B^
Lysine	15.91 a	18.15 B	30.41 ^B^	41.33 b	35.01 C	33.27 ^B^	8.72 a	9.42 A	12.12 ^A^	7.54 a	9.85 A	13.67 ^A^
Methionine	10.16 b	12.77 C	15.06 ^A^	11.28 b	24.84 B	33.87 ^B^	4.78 a	6.04 A	9.14 ^A^	7.37 ab	7.80 A	9.20 ^A^
Phenylalanine	3.35 a	26.32 A	41.65 ^A^	47.47 c	57.44 C	64.98 ^B^	35.98 b	36.35 B	37.26 ^A^	10.41 a	18.57 A	30.18 ^A^
Proline	24.40 a	37.74 AB	38.01 ^AB^	101.52 b	122.77 C	130.96 ^C^	37.90 a	41.96 B	46.11 ^B^	23.93 a	25.79 A	27.07 ^A^
Serine	ND a	ND A	ND ^A^	7.61 b	14.27 C	30.29 ^C^	8.22 b	8.03 B	12.93 ^B^	ND a	ND A	ND ^A^
Threonine	ND	ND	ND	ND	ND	ND	ND	ND	ND	ND	ND	ND
Tryptophan	ND a	ND A	20.89 ^B^	24.02 bc	21.96 B	ND ^A^	24.91 c	ND A	ND ^A^	18.93 b	21.36 B	24.99 ^B^
Tyrosine	ND a	ND A	ND ^A^	34.46 c	50.52 C	ND ^A^	25.63 b	32.67 B	34.81 ^B^	ND a	ND A	ND ^A^
Valine	30.04 a	36.32 A	34.27 ^A^	52.02 b	90.87 C	107.30 ^C^	48.31 b	54.27 B	65.31 ^B^	21.95 a	43.08 AB	66.06 ^B^
Cystine	ND	ND	ND	ND	ND	ND	ND	ND	ND	ND	ND	ND
Taurine	13.58 a	29.50 BC	49.87 ^C^	10.32 a	20.48 AB	22.58 ^B^	13.03 a	11.07 A	11.12 ^A^	28.91 b	34.28 C	49.76 ^C^
Citrulline	21.28 a	25.93 A	29.73 ^A^	33.27 b	27.56 A	24.61 ^A^	35.34 b	31.40 A	28.88 ^A^	22.27 a	25.32 A	26.85 ^A^
Sarcosine	32.78 b	ND A	ND ^A^	42.33 b	35.59 B	ND ^A^	ND a	ND A	23.43 ^B^	35.26 b	40.85 C	46.43 ^C^
Ornithine	ND a	33.36 A	37.72 ^A^	52.63 b	54.09 C	67.81 ^B^	ND a	44.18 B	45.25 ^A^	ND a	32.83 A	35.86 ^A^
Total free amino acids (TFAA)	447.94 a ± 16.08	574.22 A ± 9.36	800.92 ^B^ ± 28.63	982.13 b ± 46.77	1218.51 B ± 40.13	1592.78 ^D^ ± 19.06	1099.52 c ± 21.71	1183.61 B ± 8.31	1437.59 ^C^ ± 38.59	421.21 a ± 4.22	535.39 A ±16.10	702.30 ^A^ ± 18.39

ND—not detected; means ± standard deviations; a–d—means with different letters in line for 1 week of ripening are significantly different (*p* < 0.05, n = 6); A–D—means with different letters in line for 3 weeks of ripening are significantly different (*p* < 0.05, n = 6); ^A–D^—means with different superscript letters in line for 5 weeks of ripening are significantly different (*p* < 0.05, *n* = 6).

**Table 5 materials-13-01835-t005:** Concentrations of biogenic amines (BAs) in Dutch-type cheese models during ripening (mg∙kg^−1^).

Added	Control			LC 11454			LC 2379			LC 476		
Time of Ripening (week) Biogenic Amines	1	3	5	1	3	5	1	3	5	1	3	5
Agmatine	ND	ND	ND	ND	ND	ND	ND	ND	ND	ND	ND	ND
Phenylethylamine	ND	ND	ND	ND	ND	ND	ND	ND	ND	ND	ND	ND
Histamine	17.69 a	22.33 A	40.98 ^B^	27.51 b	33.79 B	42.29 ^B^	14.98 a	35.11 B	37.10 ^B^	17.58 a	21.67 A	24.34 ^A^
Cadaverine	0.64 b	0.78 C	4.62 ^B^	0.51 ab	0.43 AB	0.60 ^A^	0.64 b	0.29 A	0.13 ^A^	0.43 a	0.55 B	1.13 ^A^
Putrescine	0.32 c	0.38 AB	3.05 ^B^	0.19 bc	0.11 A	ND ^A^	0.12 ab	0.16 AB	ND ^A^	ND a	0.56 C	0.14 ^A^
Spermidine	1.62 c	1.17 B	0.48 ^B^	0.51 ab	0.26 A	ND^A^	0.62 b	0.86 B	1.03 ^C^	0.34 a	0.83 B	1.07 ^C^
Spermine	1.60 d	0.31 A	ND ^A^	0.39 c	0.16 A	ND ^A^	0.22 b	0.22 A	ND ^A^	ND a	0.21 A	0.31 ^B^
Tryptamine	2.49 b	3.88 B	ND ^A^	ND a	ND A	ND ^A^	4.39 c	3.64 B	3.22 ^B^	ND a	3.63 B	4.59 ^C^
Tyramine	1.21 c	1.32 B	4.29 ^C^	0.95 b	1.23 B	1.42 ^B^	0.92 b	0.70 A	0.65 ^A^	0.40 a	0.67 A	0.90 ^A^
Total BA	25.56 bc ± 1.13	30.17 AB ± 3.19	53.41 ^C^ ± 4.78	30.05 c ± 0.34	35.97 BC ± 0.30	44.31 ^B^ ± 0.20	21.90 ab ± 0.12	40.99 C ± 2.20	42.14 ^B^ ± 4.52	18.76 a ± 1.49	28.10 A ± 1.58	32.48 ^A^ ± 0.94

ND—not detected; means ± standard deviations; a–d—means with different letters in line for 1 week of ripening are significantly different (*p* < 0.05, n = 6); A–D—means with different letters in line for 3 weeks of ripening are significantly different (*p* < 0.05, *n* = 6); ^A–D^—means with different superscript letters in line for 5 weeks of ripening are significantly different (*p* < 0.05, *n* = 6).

**Table 6 materials-13-01835-t006:** Concentrations of bioactive peptides (BP) in Dutch-type cheese models during ripening (mg∙kg^−1^).

Added	Control			LC 11454			LC 2379			LC 476		
Time of Ripening (week) Bioactive Peptides	1	3	5	1	3	5	1	3	5	1	3	5
Anserine	66.33 a	85.17 A	79.29 ^A^	88.63 c	95.95 B	106.92 ^B^	103.08 d	94.10 B	78.55 ^A^	79.35 b	80.01 A	79.23 ^A^
L-carnosine	8.37 b	ND A	ND ^A^	ND a	ND A	35.23 ^B^	29.83 c	11.48 B	ND ^A^	9.99 b	ND A	ND ^A^
Total BP	74.70 a ± 4.87	85.17 A ± 2.98	79.29 ^A^ ± 2.62	88.63 b ± 1.33	95.95 B ± 2.63	142.15 ^B^ ± 4.00	132.91 c ± 3.82	105.58 C ± 5.37	78.55 ^A^ ± 8.95	89.34 b ± 4.00	80.01 A ± 1.73	79.23 ^A^ ± 1.09

ND—not detected; means ± standard deviations; a–d—means with different letters in line for 1 week of ripening are significantly different (*p* < 0.05, n = 6); A–D—means with different letters in line for 3 weeks of ripening are significantly different (*p* < 0.05, n = 6); ^A–D^—means with different superscript letters in line for 5 weeks of ripening are significantly different (*p* < 0.05, *n* = 6).

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
