# Peer review of "Impact of Nisin-Producing Strains of Lactococcus lactis on the Contents of Bioactive Dipeptides, Free Amino Acids, and Biogenic Amines in Dutch-Type Cheese Models"

_materials, 2020, doi:10.3390/ma13081835_

Round 1

Author Response

Dear Reviewer 1,

Thank you very much for reviewing our manuscript: Impact of nisin-producing strains of Lactococcus lactis on the bioactive dipeptides, free amino acids, and biogenic amines contents in Dutch-type cheese models. We have adopted all your suggestions.

Your suggestions have seriously contributed to the improvement of our manuscript. All changes compared to the original version have been highlighted in yellow. Hope the revised manuscript will be evaluated as improved, in any case, we are willing to consider any further request.

The study is interesting but, the chemical composition of the cheeses would explain better your results. Please include the data of the chemical composition of cheeses.

The data of the cheese models chemical composition was included.

Did you use the same cheese wheel (or cheese mass) for sampling throughout ripening?

Samples were collected from each of the three replicates for each cheese model variant during ripening.

  1. Introduction

- Line 33: …EFSA (2011)…. Please add to the references. Is it the same with ref. 47?

Corrected in modified text.

  1. Material and Methods

- Line 121: … Schleifer et al …Is it a reference?

Corrected in modified text.

- Lines 136-137: ‘The use of …milk’ This is not clear

Corrected in modified text

- Lines 147-149: Correct to ..at 320 g …at 1400 g

Corrected in modified text.

  1. Results and discussion

- Line 264: Glutamic acid, Valine and Histidine were much increased during ripening of cheeses LC 11454. How do you explain it?

Amino acids in cheese are the result of microbial metabolic activities in which some amino acids are used and others may be produced and excreted in to the cheese matrix.

In many cheese variety during ripening, the content of the single FAA increased. It indicates that the addition some of LABs effectuate some important changes in amino acid composition in cheese. Histidine, Glutamic acid increased as a results of adding LC 11454.

Val were found in large amounts in long-ripened Caciocavallo cheese and in other cheeses such as Idiazàbal Picante, Serra da Estrela, Fiore Sardo, Teleme, goat cheese, Italian cheese. Val with the other branched chain AA (Ile), the aromatic AA (Phe, Tyr), and Met, are the main precursors of key aroma compounds.

- Line 268: Proline was much higher in cheeses LC 11454 from the beginning, compared to other cheeses. How do you explain it?

Casein contain proportionally large amount of Pro and to release the amino acids needed by the bacteria Pro has to be released from the peptide chain. While using a Pep-X like enzymes one Pro is released in cheese for every other amino acid that may be used by bacteria and such an overproduction could explain a large amount of Pro.

Lysine and proline are biosynthetic products from glutamate and arginine metabolism. These amino acids are important components of cell wall structures in their D-form. The L-forms are still catabolized to produce energy.

Notably, metabolic analysis shows that glutamate participates in over 150 metabolic pathways in LAB. Glutamine is interconvertible with glutamate by glutamine synthase and an ATase, which acts as a nitrogen-fixing mechanism. In our study, only in cheese model with additional LC 11454 glutamine content was found throughout the entire ripening period which could result in the conversion of glutamate to proline and explain its high content from the beginning of their ripening.

- Line 295 -297: Which was the proteolytic microbiota in your cheeses? If it was the adjunct starters, please make it clearer.

Corrected in modified text.

- Line 302: Please correct to ‘…in the cheese models, although both, control cheese and LC11454 cheese, which had the higher amount of histidine showed the higher amount of histamine too’.

Corrected in modified text.

- Lines 309-310: The truth is that the chemical composition of the cheeses would explain better your results.

We agree with this so we included the chemical composition in our manuscript.

- Line 315: NSLAB in your cheeses is limited, since you used microfiltered and pasteurized (74oC/15s) milk and sterilized bottles for cheesemaking.

Corrected in modified text.

- Line 318-319: How do you explain that control cheese appeared tyramine whereas it did not contain tyrosine?

The amines are synthesized by lactic acid bacteria through the decarboxylation of amino acids present in the food matrix. However, the concentration of biogenic amines in fermented foodstuffs is influenced by many environmental factors. In addition, it is now known that peptides containing amino acids precursors of biogenic amines can be used by bacteria to produce these biogenic amines. For example, tyramine can be produced from peptides containing tyrosine and free tyrosine is not the only precursor for tyramine production. In general tyramine is produced in foods from the natural breakdown of the amino acid tyrosine or tyrosine containing peptides.

Reviewer 2 Report

Dear author the manuscript "Impact of nisin-producing strains of Lactococcus lactis on the bioactive dipeptides, free amino acids, and biogenic amines contents in Dutch-type cheese models" reports novel and relevent inputs on cheese production. 

the minor concerns I have is related to the statistical analysis performed: the authors chose to apply one way anova and to compare all groups with eachothers. did the authors consider to apply also a repeted measure anova as the monitored the ripening over time? or a mixed model? 

Also, what is the rationale behind the differences in aa and amines contents at week 1?  can be relevant to measure the starting amount? at time 0 (before ripening)?

kind regards

Author Response

Dear Reviewer 2,

Thank you very much for reviewing our manuscript: Impact of nisin-producing strains of Lactococcus lactis on the bioactive dipeptides, free amino acids, and biogenic amines contents in Dutch-type cheese models.

Your suggestions have seriously contributed to the improvement of our manuscript. All changes compared to the original version have been highlighted in yellow. Hope the revised manuscript will be evaluated as improved, in any case, we are willing to consider any further request.

The minor concerns I have is related to the statistical analysis performed: the authors chose to apply one way anova and to compare all groups with eachothers. did the authors consider to apply also a repeted measure anova as the monitored the ripening over time? or a mixed model? 

We considered the analysis of variance in the content of aa and biogenic amines during ripening. However, the application of subsequent letters about the significance of differences would make Tables 1 and 2 difficult to read.

Also, what is the rationale behind the differences in aa and amines contents at week 1?  can be relevant to measure the starting amount? at time 0 (before ripening)?

We agree that it would be better to include the results of aa and biogenic amines at time zero, now we see it.

Reviewer 3 Report

Materials Manuscript ID 756930

General comments

In this manuscript there are both points of interest and originality and some weaknesses. The latter are detailed as specific comments and need to be carefully addressed.

Specific comments

Introduction, line 33: please add reference number (47).

Introduction, line 36: Most the food products? English revision required.

Introduction, lines 56-57: this sentence seems to be useless, because doesn’t add any new concept in comparison with lines 50-55. Please remove or rewrite.

Introduction, line 60: not lysine, but nisin.

Introduction, line 62: comma after Veneri has to be removed.

Introduction, line 107: ripening? Ripened. Probably you can say simply “in cheeses” rather than “in ripened rennet cheeses”. Data in cheeses in general are lacking. Moreover, considering that alanine and hystidine are precursors of the studied dipeptides, it could be of interest to add a sentence about the content of these 2 amino acids in cheeses.

Materials and Methods 2.1, lines 116-121: please add, after “used as adjunct cultures”, “one for each of 3 cheese variants” or similar.

Materials and Methods 2.1, line 118: please specify that L. lactis PCM 476 is not a nisin-producing strain. This is not clear for the reader until Results and Discussion section has been read.

Materials and Methods 2.1, line 126: “were”, not “was”.

Materials and Methods 2.1, lines 127-129: The authors say that they selected the additional cultures “based on their good technological aptitude and proteolytic activity”. How they tested these activities and what are main results? This is important particularly in relation to proteolytic activity, since the used adjunct cultures could have determined their effect in cheese models, both by nisin activity with consequent SLAB lysis and directly by production of their proteolytic enzymes.

Materials and Methods 2.1, lines 129-130: It is not correct to put this sentence in MM section. You can use it in the Discussion.

Materials and Methods 2.2, lines 133-137: I don’t agree with the choice of the authors to use microfiltered pasteurized milk. Using pasteurized not microfiltered milk (as, for example, Hines et al., that the authors cite as the reference for their cheese models) should have allowed to maintain a more or less limited number of NSLAB (mesophilic lactobacilli, enterococco…), as it usually happens in cheese-making processes. It is true that in some cases the cheese-making industry uses microfiltered milk, for instance to limit late blowing…

Thus, please, clarify the reason for your choice (the sentence in lines 136-137 is not clear at all and not correct: sterilized milk?? Microfiltered pasteurized is not a sterilized milk): to exclude NSLAB and study only the interaction SLAb/adjunct cultures?

Materials and Methods 2.2, line 139: Fromase: specify type of coagulant (microbial coagulant) and possibly IMCU activity.

Materials and Methods 2.2, line 155: remove “(descrive below)”, is superfluous.

Materials and Methods 2.2, lines 158-164: remove “and a coagulant”, is superfluous. Change from “consisting of…” to “made with milk inoculated with…” or similar.

Materials and Methods 2.3: can you specify LOQ and LOD?

Materials and Methods 2.3, line 170: comma after Roszko has to be removed.

Materials and Methods 2.3, line 170: change to “Ca. 1.2 g” to “A quantity of 1.2 g”.

Materials and Methods 2.3, lines 178-179: remove “there were added” and add “were added” after (Sigma Aldrich, Poland).

Materials and Methods 2.3, line 188: change from “flaks” to “flask”.

Materials and Methods 2.3, line 205: “this”, not “these”; “analyses”, not “analysis”.

Results and Discussion 3.1, line 224: “is”, not “are”.

Results and Discussion 3.1, line 259-260: this sentence is not correct: Diana et al (24) reported an ornithine content range of 0.40-1.08 g/kg, that is higher than that you obtained. Renes et al, 2019 (reference n. 25) obtained about 2000 mg/kg. Please, reformulate the sentence in a more precise way.

Results and Discussion 3.2, lines 294-297: “Differences were also noted in the other cheese models…”: in other studies? In reference 10? This sentence is not clear. Please, reformulate.

Results and Discussion 3.2, lines 298-300: It’s not a decrease but a lower increase in comparison to control.

Results and Discussion 3.2, lines 308-309: “These results confirm dependencies described by other authors…”: please, explain why.

Results and Discussion 3.2, lines 313-315: the same as lines 298-300, please, reformulate.

Results and Discussion 3.2, lines 318: please, check accuracy of this sentence about statistical significance  of histamine values in control and Lc 11454 cheeses in comparison to the 2 other variants at the different times.

Results and Discussion 3.2, lines 345-346: with regard to total BA content, the sentence “was found a lower content compared to the contro cheese model” is correct after 5 weeks, not in all cases considering other ripening times.

Results and Discussion 3.2, line 350, as in MM: not “microfiltrated” but “microfiltered”.

Results and Discussion 3.3, lines 361-362: with the aim of better explain a possible additional health effect in cheese, please, add, as an example, some information about values of carnosine and anserine in meat samples, where their content is well known (see, for example, your reference n. 29, Mori et al.). This is not to diminish your results, but to better discuss them.

Author Response

Dear Reviewer 3

Thank you very much for reviewing our manuscript: Impact of nisin-producing strains of Lactococcus lactis on the bioactive dipeptides, free amino acids, and biogenic amines contents in Dutch-type cheese models. We have adopted all your suggestions.

Your suggestions have seriously contributed to the improvement of our manuscript. All changes compared to the original version have been highlighted in yellow. Hope the revised manuscript will be evaluated as improved, in any case, we are willing to consider any further request.

Introduction, line 33: please add reference number (47).

Corrected in modified text.

Introduction, line 36: Most the food products? English revision required.

Corrected in modified text.

Introduction, lines 56-57: this sentence seems to be useless, because doesn’t add any new concept in comparison with lines 50-55. Please remove or rewrite.

Corrected in modified text.

Introduction, line 60: not lysine, but nisin.

Corrected in modified text.

Introduction, line 62: comma after Veneri has to be removed.

Corrected in modified text.

Introduction, line 107: ripening? Ripened. Probably you can say simply “in cheeses” rather than “in ripened rennet cheeses”. Data in cheeses in general are lacking. Moreover, considering that alanine and hystidine are precursors of the studied dipeptides, it could be of interest to add a sentence about the content of these 2 amino acids in cheeses.

Corrected in modified text.

Materials and Methods 2.1, lines 116-121: please add, after “used as adjunct cultures”, “one for each of 3 cheese variants” or similar.

Corrected in modified text.

Materials and Methods 2.1, line 118: please specify that L. lactis PCM 476 is not a nisin-producing strain. This is not clear for the reader until Results and Discussion section has been read.

Corrected in modified text.

Materials and Methods 2.1, line 126: “were”, not “was”.

Corrected in modified text.

Materials and Methods 2.1, lines 127-129: The authors say that they selected the additional cultures “based on their good technological aptitude and proteolytic activity”. How they tested these activities and what are main results? This is important particularly in relation to proteolytic activity, since the used adjunct cultures could have determined their effect in cheese models, both by nisin activity with consequent SLAB lysis and directly by production of their proteolytic enzymes.

We have not tested activities of additional strains of Lactococcus bacteria used in the study. We selected them on the basis of information obtained from the Polish Collection of Microorganisms of the Ludwik Hirszfeld Institute of Immunology and Experimental Therapy of the Polish Academy of Sciences, from which we received data on their technological usefulness, proteolytic activity and the ability or not to produce nisin. After obtaining such information based on them, we selected the strains used in the study. In the case of the ATCC 11454 strain, which we had in our own resources, we did not have data on its technological usefulness, but we knew that it is capable of producing nisin.

Materials and Methods 2.1, lines 129-130: It is not correct to put this sentence in MM section. You can use it in the Discussion.

Corrected in modified text.

Materials and Methods 2.2, lines 133-137: I don’t agree with the choice of the authors to use microfiltered pasteurized milk. Using pasteurized not microfiltered milk (as, for example, Hines et al., that the authors cite as the reference for their cheese models) should have allowed to maintain a more or less limited number of NSLAB (mesophilic lactobacilli, enterococco…), as it usually happens in cheese-making processes. It is true that in some cases the cheese-making industry uses microfiltered milk, for instance to limit late blowing…

Microfiltered milk was used to obtain cheese models for three reasons:

  1. it is milk that has subjected to cold sterilization to eliminate the effect of NSLAB on the obtained cheese models
  2. microfiltered milk used in this study was subjected to “mild” pasteurization at 74 °C / 15s (HTST), which did not hinder the coagulating enzyme action. Other commercially available milk in Poland, most often pasteurized at 76-85 °C / 15s, without explicit manufacturer's declaration which temperature it has undergone.
  3. in the production of Dutch type cheeses in Polish factories, the use of bactofugation or microfiltration to prevent late blowing is common because the use of other alternative methods to reduce late blowing has many disadvantages. Of this two methods of remove microflora from raw milk, microfiltration is better than bactofugation.

Thus, please, clarify the reason for your choice (the sentence in lines 136-137 is not clear at all and not correct: sterilized milk?? Microfiltered pasteurized is not a sterilized milk): to exclude NSLAB and study only the interaction SLAb/adjunct cultures?

You are right, this is not sterilized milk, but from a microbiological point of view, microfiltered pasteurized milk is close to sterilized milk. Practically only bacteria derived from reinfection after pasteurization can be present in such cheese milk possibly. We can only add that we have checked this milk many times in terms of the number of microorganisms and it always showed not CFU in 0.1 ml. In addition, this milk was subjected to storage tests at 6 ° C and showed a shelf life of up to 2 months, despite the manufacturer's declaration of only 21 days at 6 ° C. Yes you are right we meant to eliminate of possible impact of NSLAB on results.

Materials and Methods 2.2, line 139: Fromase: specify type of coagulant (microbial coagulant) and possibly IMCU activity.

Corrected in modified text.

Materials and Methods 2.2, line 155: remove “(descrive below)”, is superfluous.

Corrected in modified text.

Materials and Methods 2.2, lines 158-164: remove “and a coagulant”, is superfluous. Change from “consisting of…” to “made with milk inoculated with…” or similar.

Corrected in modified text.

Materials and Methods 2.3: can you specify LOQ and LOD?

LOQ in 5–10 ngmL-1 range, LOD in 10 ngmL-1

Materials and Methods 2.3, line 170: comma after Roszko has to be removed.

Corrected in modified text.

Materials and Methods 2.3, line 170: change to “Ca. 1.2 g” to “A quantity of 1.2 g”.

Corrected in modified text.

Materials and Methods 2.3, lines 178-179: remove “there were added” and add “were added” after (Sigma Aldrich, Poland).

Corrected in modified text.

Materials and Methods 2.3, line 188: change from “flaks” to “flask”.

Corrected in modified text.

Materials and Methods 2.3, line 205: “this”, not “these”; “analyses”, not “analysis”.

 Corrected in modified text.

Results and Discussion 3.1, line 224: “is”, not “are”.

Corrected in modified text.

Results and Discussion 3.1, line 259-260: this sentence is not correct: Diana et al (24) reported an ornithine content range of 0.40-1.08 g/kg, that is higher than that you obtained. Renes et al, 2019 (reference n. 25) obtained about 2000 mg/kg. Please, reformulate the sentence in a more precise way.

Corrected in modified text.

Results and Discussion 3.2, lines 294-297: “Differences were also noted in the other cheese models…”: in other studies? In reference 10? This sentence is not clear. Please, reformulate.

Corrected in modified text.

Results and Discussion 3.2, lines 298-300: It’s not a decrease but a lower increase in comparison to control.

Corrected in modified text.

Results and Discussion 3.2, lines 308-309: “These results confirm dependencies described by other authors…”: please, explain why.

Corrected in modified text.

Results and Discussion 3.2, lines 313-315: the same as lines 298-300, please, reformulate.

Corrected in modified text.

Results and Discussion 3.2, lines 318: please, check accuracy of this sentence about statistical significance of histamine values in control and Lc 11454 cheeses in comparison to the 2 other variants at the different times.

Corrected in modified text.

Results and Discussion 3.2, lines 345-346: with regard to total BA content, the sentence “was found a lower content compared to the control cheese model” is correct after 5 weeks, not in all cases considering other ripening times.

Corrected in modified text.

Results and Discussion 3.2, line 350, as in MM: not “microfiltrated” but “microfiltered”.

Corrected in modified text.

Results and Discussion 3.3, lines 361-362: with the aim of better explain a possible additional health effect in cheese, please, add, as an example, some information about values of carnosine and anserine in meat samples, where their content is well known (see, for example, your reference n. 29, Mori et al.). This is not to diminish your results, but to better discuss them.

Corrected in modified text.

Reviewer 4 Report

To the Authors of the manuscript entitled “Impact of nisin-producing strains of Lactococcus lactis on the bioactive dipeptides, free amino acids, and biogenic amines contents in Dutch-type cheese models”.

After carefully reading and reviewing the manuscript. In my opinion. the methods you described and the results are consistent. The discussion of the results is consistent with the results obtained by the authors and previous research work.  There are some changes/edits I suggest need to be made, most of them related to English grammar. My major concern is that this manuscript is not be within the scope of this particular journal (materials science and engineering). I would suggest that other journals published by MDPI such as Foods, Fermentation or eve Nutrients are more appropriate for this type of experimental work. Please, read below for a list of comments and suggestions.

Best regards,

  1. I suggest the authors change “cheese models” for “model cheeses” or “model cheese systems” throughout the entire manuscript
  2. Line 46. Change “symbol” for “code” or “European Union code”
  3. Line 76. The statement about “nutritive functions” seems like a mere speculation. Authors should edit it for clarity and/or provide a reference. Maybe give examples of such potential beneficial functions?
  4. Line 85. Remove e.g.
  5. Line 136. Reads: “….hardle methods”. Is that a typo? Should it be “hurdle”? Explain, and if necessary correct
  6. Line 148. Reads: “…possibly the greatest amount of….”. Should it be: “the largest possible amount….”?
  7. Line 150. Define “miniature”, give an approximate mass for those model cheeses
  8. Lines 157-165. Create a table to summarize the composition of the different experimental trials. It is more informative and easier to read
  9. Line 165. Reads: “were obtained in three times replicates”. Should say “were prepared in triplicate”
  10. Line 189. There is no need to use capital letters for the entire name of the rotary evaporator (Büchi Labortechnik AG)
  11. Lines 199-202. Make a table to show the composition of the mobile phase (gradient)
  12. Line 205. Should say: “analyses were done in duplicate”
  13. Lines 320-322. Authors should give examples of the undesirable and potentially toxic effect of such biogenic amines. What kind of symptoms are developed by humans after exposure to BA?
  14. Table 2. LC476. Erase the letter w next to 1,3,5
  15. Line 342. Edit sentence, review the grammar. Should it say “observed that, when adding a nisin-producing strain L lactis to Minhas cheese… the content of 2-phenylethylamine after 60 days of aging was four times lower than…the control”?
  16. Lines 352-353. Add to the discussion data to explain how long does it typically take for the BA concentration to be appreciable or cause toxicity. It has already been established that BA are present in certain aged cheeses around the world.
  17. Line 385. Reads: “…a health-promoting compounds…”. Remove the word “a”

Author Response

 Dear Reviewer 4,

Thank you very much for reviewing our manuscript: Impact of nisin-producing strains of Lactococcus lactis on the bioactive dipeptides, free amino acids, and biogenic amines contents in Dutch-type cheese models. We have adopted all your suggestions.

Your suggestions have seriously contributed to the improvement of our manuscript. All changes compared to the original version have been highlighted in yellow. Hope the revised manuscript will be evaluated as improved, in any case, we are willing to consider any further request.

  1. I suggest the authors change “cheese models” for “model cheeses” or “model cheese systems” throughout the entire manuscript

Cheese model is something that is not a specific cheese but only imitates conditions similar to cheese (base of cheese).

  1. Line 46. Change “symbol” for “code” or “European Union code”

Corrected in modified text.

  1. Line 76. The statement about “nutritive functions” seems like a mere speculation. Authors should edit it for clarity and/or provide a reference. Maybe give examples of such potential beneficial functions?

Corrected in modified text.

  1. Line 85. Remove e.g.

Corrected in modified text.

  1. Line 136. Reads: “….hardle methods”. Is that a typo? Should it be “hurdle”? Explain, and if necessary correct

We decided to delete this sentence.

  1. Line 148. Reads: “…possibly the greatest amount of….”. Should it be: “the largest possible amount….”?

Corrected in modified text.

  1. Line 150. Define “miniature”, give an approximate mass for those model cheeses

Corrected in modified text.

  1. Lines 157-165. Create a table to summarize the composition of the different experimental trials. It is more informative and easier to read

Corrected in modified text.

  1. Line 165. Reads: “were obtained in three times replicates”. Should say “were prepared in triplicate”

Corrected in modified text.

  1. Line 189. There is no need to use capital letters for the entire name of the rotary evaporator (Büchi Labortechnik AG)

Corrected in modified text.

  1. Lines 199-202. Make a table to show the composition of the mobile phase (gradient)

Corrected in modified text.

  1. Line 205. Should say: “analyses were done in duplicate”

Corrected in modified text.

  1. Lines 320-322. Authors should give examples of the undesirable and potentially toxic effect of such biogenic amines. What kind of symptoms are developed by humans after exposure to BA?

Corrected in modified text.

  1. Table 2. LC476. Erase the letter w next to 1,3,5

Corrected in modified text.

  1. Line 342. Edit sentence, review the grammar. Should it say “observed that, when adding a nisin-producing strain L lactis to Minhas cheese… the content of 2-phenylethylamine after 60 days of aging was four times lower than…the control”?

Corrected in modified text.

  1. Lines 352-353. Add to the discussion data to explain how long does it typically take for the BA concentration to be appreciable or cause toxicity. It has already been established that BA are present in certain aged cheeses around the world.

Corrected in modified text.

  1. Line 385. Reads: “…a health-promoting compounds…”. Remove the word “a”

Corrected in modified text.

Round 2

Reviewer 3 Report

Materials Manuscript ID 756930

Second revision

General comments

The authors addressed most of the issues and the paper has been improved. Some other minor issues are presented as "Specific comments".

Probably due to the limited time available for the revisions, changes have been sloppily written, with many mispellings and similar. English must be revised.

Specific comments

Materials and Methods 2.2, lines 131-132: I suggest to change from “to eliminate all microflora of raw milk” (that is not correct in a not sterilized milk) to “to minimize the presence of microorganisms from raw milk and particularly NSLAB” or similar.

Materials and Methods 2.2, line 155: change from “demonstrated” to “specified” or “shown”.

Materials and Methods 2.4: the authors answered to my question about LOD and LOQ “LOQ in the range 5-10ng/ml, LOD 10 ng/ml”. Firstly, I guess is ng/mg and not ng/ml, since is a solid matrix. Then, considering that LOQ has, at the best, the same value of LOD, not a lesser one, probably is: LOD about 5 ng/mg and LOQ about 10 ng/mg…Not fantastic in comparison with many other papers, but acceptable.

Results and Discussion 3.1, line 206: not Table 1 but Table 3.

Results and Discussion 3.1, lines 210 and 214: remove (Table 3), is superfluous.

Results and Discussion 3.2, line 296: change from “started” to “starter”.

Results and Discussion 3.3, lines 346-358: when the authors discuss the BAs content of their cheese in comparison with other cheeses, they have to highlight, to be correct, that, in their model cheeses, they chose to minimize the presence of microorganisms relevant for BAs production in cheeses, NSLAB in primis, and their content is exclusively determined by the specific starter and adjunct cultures used. The same for the comparison with Minas cheese, lines 359-363.

Results and Discussion 3.4, lines 390-395: The authors added values of carnosine and anserine, as requested. To better explain the quantities found in cheese, I suggest to further add a sentence like this: “It is unquestionable that the recorded levels are low in comparison of those of other foods. As regards this, Mori et al…” in line 390. Or similar.

Conclusions: line 414: change “contribute” to “might contribute”.

Author Response

Dear reviewer

Once again, we would like to thank you for your contribution to improving our manuscript.

We hope that revised manuscript will be evaluated as improved, in any case, we are willing to consider any further request.

Materials and Methods 2.2, lines 131-132: I suggest to change from “to eliminate all microflora of raw milk” (that is not correct in a not sterilized milk) to “to minimize the presence of microorganisms from raw milk and particularly NSLAB” or similar.

Corrected.

Materials and Methods 2.2, line 155: change from “demonstrated” to “specified” or “shown”.

Corrected.

Materials and Methods 2.4: the authors answered to my question about LOD and LOQ “LOQ in the range 5-10ng/ml, LOD 10 ng/ml”. Firstly, I guess is ng/mg and not ng/ml, since is a solid matrix. Then, considering that LOQ has, at the best, the same value of LOD, not a lesser one, probably is: LOD about 5 ng/mg and LOQ about 10 ng/mg…Not fantastic in comparison with many other papers, but acceptable.

Yes, it was a typo.

Results and Discussion 3.1, line 206: not Table 1 but Table 3.

Corrected.

Results and Discussion 3.1, lines 210 and 214: remove (Table 3), is superfluous.

Corrected.

Results and Discussion 3.2, line 296: change from “started” to “starter”.

Corrected.

Results and Discussion 3.3, lines 346-358: when the authors discuss the BAs content of their cheese in comparison with other cheeses, they have to highlight, to be correct, that, in their model cheeses, they chose to minimize the presence of microorganisms relevant for BAs production in cheeses, NSLAB in primis, and their content is exclusively determined by the specific starter and adjunct cultures used. The same for the comparison with Minas cheese, lines 359-363.

Corrected.

Results and Discussion 3.4, lines 390-395: The authors added values of carnosine and anserine, as requested. To better explain the quantities found in cheese, I suggest to further add a sentence like this: “It is unquestionable that the recorded levels are low in comparison of those of other foods. As regards this, Mori et al…” in line 390. Or similar.

Corrected.

Conclusions: line 414: change “contribute” to “might contribute”.

Corrected.
